# Splicing Modulation Results in Aberrant Isoforms and Protein Products of p53 Pathway Genes and the Sensitization of B Cells to Non-Genotoxic MDM2 Inhibition

**DOI:** 10.3390/ijms24032410

**Published:** 2023-01-26

**Authors:** Erhan Aptullahoglu, Carmela Ciardullo, Jonathan P. Wallis, Helen Marr, Scott Marshall, Nick Bown, Elaine Willmore, John Lunec

**Affiliations:** 1Medical Faculty, Newcastle University Cancer Centre, Newcastle upon Tyne NE2 4AD, UK; 2Department of Haematology, Freeman Hospital, Newcastle upon Tyne NHS Foundation Trust, Newcastle upon Tyne NE7 7DN, UK; 3Department of Haematology, City Hospitals Sunderland NHS Trust, Sunderland SR4 7TP, UK; 4Northern Genetics Service, Institute of Genetic Medicine, Newcastle upon Tyne NE1 4EP, UK

**Keywords:** p53 pathway, splicing modulation, MDM2–p53 antagonists, RG7388 (idasanutlin), E7107

## Abstract

Several molecular subtypes of cancer are highly dependent on splicing for cell survival. There is a general interest in the therapeutic targeting of splicing by small molecules. E7107, a first-in-class spliceosome inhibitor, showed strong growth inhibitory activities against a large variety of human cancer xenografts. Chronic lymphocytic leukaemia (CLL) is a clinically heterogeneous hematologic malignancy, with approximately 90% of cases being *TP53* wild-type at diagnosis. An increasing number of studies are evaluating alternative targeted agents in CLL, including MDM2–p53 binding antagonists. In this study, we report the effect of splicing modulation on key proteins in the p53 signalling pathway, an important cell death pathway in B cells. Splicing modulation by E7107 treatment reduced full-length MDM2 production due to exon skipping, generating a consequent reciprocal p53 increase in *TP53*^WT^ cells. It was especially noteworthy that a novel p21^WAF1^ isoform with compromised cyclin-dependent kinase inhibitory activity was produced due to intron retention. E7107 synergized with the MDM2 inhibitor RG7388, via dual MDM2 inhibition; by E7107 at the transcript level and by RG7388 at the protein level, producing greater p53 stabilisation and apoptosis. This study provides evidence for a synergistic MDM2 and spliceosome inhibitor combination as a novel approach to treat CLL and potentially other haematological malignancies.

## 1. Introduction

There are many cancer-relevant genes subject to aberrant alternative splicing, indicating that dysregulation of the splicing machinery is associated with different hallmarks of the cancer phenotype, including the promotion of angiogenesis [1], the induction of cell proliferation [2,3], promoting resistance to cancer-targeted therapy or immunotherapy [4,5], or avoiding apoptosis [6]. Cancer cells carrying heterozygous mutations in splicing factor genes are preferentially dependent on wild-type spliceosome function [7,8,9]; therefore, treating these cells with pre-mRNA spliceosome inhibitors provides a potential therapeutic strategy. Multiple protein/protein and protein/RNA interactions that contribute to the accuracy of splicing can be manipulated for therapeutic purposes [10]. Over the last two decades, multiple natural compounds derived from bacteria and their synthetic analogues have been identified to inhibit or deregulate splicing [11,12,13]. Although these early examples of splicing modulators showed promising anticancer activity in various in vitro and/or in vivo studies with IC_50_ values in the low nanomolar range, they were chemically unstable and thus unsuitable for clinical use.

E7107, one of the synthetic derivatives of the natural product pladienolide, displayed strong antitumour activities against a large variety of human cancer cell lines and tumour xenografts [14,15]. In preclinical studies, E7107 induced cell cycle arrest at both G_1_ and G_2_ phases, and subsequently apoptosis [16,17]. Significant tumour regression was produced by exposure to E7107 in a range of xenograft models tested, including lung, breast, and colon carcinomas [15,18]. In addition, E7107 sensitized primary CLL cells to the BCL2 inhibitor venetoclax by targeting BCL2 family members and remodelling mitochondrial apoptotic dependencies [19].

Inhibiting the p53–MDM2 interaction with synthetic small molecules has been investigated as a therapeutic strategy for the activation of wild-type p53 [20]. One of the compounds developed is RG7388 (idasanutlin), a second-generation MDM2–p53 binding antagonist with enhanced potency (IC_50_ = 6 nM), selectivity, and bioavailability. Both oral and intravenous formulations have been developed [21]. RG7388 has been shown to activate wild-type p53, evidenced by p53 stabilisation accompanied by the induction of p21 and MDM2 expression and the induction of p53-dependent apoptosis in different cell types [22,23,24,25], including primary CLL cells [26]. It is the first MDM2–p53 binding antagonist to have progressed through phase II and entered into phase III clinical trials (ClinicalTrials.gov Identifier: NCT02545283). According to recently published early clinical trial data, idasanutlin has become a promising targeted therapy agent for single and combined use [27,28,29].

Here, we hypothesized that the altered splicing of p53 pathway genes has a contributory effect on the cytotoxicity of E7107, as p53 is a master regulator of cellular death signalling pathways. The potential for synergistic combination treatment with the spliceosome inhibitor E7107 and the MDM2–p53 antagonist RG7388 was investigated in leukaemia–lymphoma cell lines and primary CLL samples.

## 2. Results

### 2.1. B-Cell Lines and Primary CLL Samples Are Sensitive to Spliceosome Inhibition Using E7107

Six out of eight cell lines were notably more sensitive (IC_50_ < 15 nM) to the spliceosome inhibitor than the other two (IC_50_ > 60 nM). The mean IC_50_ values were relatively higher for HEL and HAL-01 cell lines (60.2 ± 2.9 and 203.5 ± 14.3 nM, respectively) (Figure 1A, Appendix A). E7107 showed concentration-dependent cytotoxicity against a Nalm-6 isogenic *SF3B1* wild-type and mutant cell line pair, independently of *SF3B1* mutation status (unpaired *t*-test, *p* = 0.63). No correlation was evident between E7107 sensitivity and *TP53* status (unpaired *t*-test, *p* = 0.52) or *SF3B1* status (unpaired *t*-test, *p* = 0.64) for the primary CLL samples tested (Figure 1A). Findings from both the B-cell line panel and primary CLL samples indicate that the toxicity of E7107 is independent of *SF3B1* and *TP53* mutational status. Notably, PBMC samples from healthy volunteers were more resistant to E7107 (IC_50_ > 300 nM) than either the cell lines or primary CLL samples (Figure 1B).

### 2.2. Splicing Modulation Results in Aberrant Splicing Isoforms and Protein Products of P53 Pathway Genes

One of the goals of this study was to test the combined use of the MDM2–p53 antagonist RG7388, a *TP53*-dependent targeted therapy agent, with the spliceosome inhibitor E7107. Although the overall single-agent cytotoxic effect of E7107 appears to be independent of *TP53* mutational status, we were interested to evaluate its mechanistic effect on key proteins in the p53 signalling pathway, an important cell death pathway in B cells, before testing its combined use with a *TP53*-dependent agent. The Western blots revealed a marked concentration-dependent decrease in MDM2, a critical negative regulator of p53 [30], in all cell lines tested. Consistent with this was a reciprocal concentration-dependent increase in p53 in *TP53*^WT^ cells. In addition, a lower molecular weight p53 isoform (~45 kDa) was seen in OCI-Ly3 cells treated with E7107, likely due to alternative splicing, whereas a concomitant novel high molecular weight p21^WAF1^ isoform (~30 kDa) was evident in *TP53*^WT^ cells (Figure 1C). p21^WAF1^ is an important downstream target of p53 and plays key central roles in the negative regulation of the cell division cycle [31,32].

The RT-PCR of *MDM2* transcripts spanning exons 1 to 11 revealed an altered *MDM2* transcript length of approximately 500 bp in OCI-Ly3 cells treated with 50 nM E7107 instead of the expected full-length sequence of 1303 bp (Figure 1D). The constitutive and altered products were confirmed by Sanger sequencing to be the result of skipping exons 3 to 10 (Figure 1E, Appendix A). The RT-PCR of *CDKN1A* (p21^WAF1^) transcript with primers spanning the end of exon 3 and beginning of exon 4 revealed the presence of an additional longer transcript product of approximately 1200 bp in OCI-Ly3 cells treated with E7107, which was longer than the RT-PCR product size of 84 bp expected from the processed transcript with the chosen primers (Figure 1F). Sequencing of the longer product, purified from agarose gel, confirmed that the abnormal transcript resulted from retention of the intron between exons 3 and 4 of the *CDKN1A* (p21^WAF1^) gene (Figure 1G, Appendix A), which was consistent with the abnormal higher molecular weight protein detected by Western blot (Figure 1C).

### 2.3. The PCNA Binding Site and NLS Are Disrupted in the Intron-Retaining Isoform of P21^WAF1^

Analysis of the translated amino acid sequences and motifs of wild-type p21^WAF1^ and the intron-retaining isoform (p21^L^) revealed that despite retaining the N-terminal CDK binding site, the proliferating cell nuclear antigen (PCNA) interaction site (Figure 1H) and nuclear localization signal (NLS) domain were partially lost in the intron-retained isoform of *CDKN1A* encoding the p21^L^ protein product (Figure 2). These findings suggested that the alteration in the nuclear localization signal of p21^L^ would prevent its access to the nucleus, resulting in the loss of both CDK and PCNA inhibitory activity in the cellular context.

The expression of the p21^L^ product (Appendix A) did not affect the growth rate of Nalm-6 cells (Figure 3A). The p21^L^ intron-retained form was also examined to see if it still prevented RB phosphorylation by cyclin/CDK complexes. Western immunoblots were probed with three different phosphorylated-RB (pRB) antibodies detecting different phosphorylation sites on pRB. The MDM2 inhibitor RG7388 was used for endogenous wild-type p21 induction. The result showed that exogenous p21^L^ expression had no impact on pRB phosphorylation and hence no CDK inhibitory activity in the transduced Nalm-6 cells, in contrast to the clear suppression of RB phosphorylation seen on induction of endogenous p21^WAF1^ by RG7388 (Figure 3B).

To provide a detailed characterization of the intracellular localization of the high molecular weight p21^L^ protein, particularly to test the hypothesis that the p21^L^ protein was unable to localise to the nucleus, the subcellular distribution of p21^L^ was examined in Nalm-6–p21^L^ transduced cells by immunofluorescence. In the Nalm-6–p21^L^ transduced cells induced using 0.5 µg/mL doxycycline for 24 h to express high molecular weight p21^L^ protein, the p21 signal was detected only in the cytoplasm of the cells. Treatment with the MDM2 inhibitor RG7388 (1 µM; 24 h), however, induced endogenous p21^WAF1^ expression and the p21^WAF1^ signal was detected in both cytoplasmic and nuclear compartments (Figure 3C).

### 2.4. E7107 Sensitizes B-Cell Lines to Non-Genotoxic MDM2 Inhibition with RG7388

RG7388 treatment induced selective cytotoxicity in a panel of TP53^WT^ and TP53^MUT^ B-cell lines, consistent with its mechanism of action (Figure 4A). The combination treatment was encouraged by the observed disruption of the p53 pathway proteins by spliceosome inhibition. The growth inhibition and cytotoxic effect against B-cell lines during a 72 h exposure to RG7388 and E7107 were determined as single agents, and in combination at five equipotent concentrations between 0.25× and 4× of their respective IC_50_ concentrations (Appendix A). The combination effect of RG7388 with E7107 varied from synergism with OCI-Ly3 to moderate to slight synergism for Nalm-6 cells depending on drug concentration (Figure 4B,C). Combination treatments in both cell lines led to greater levels of p53 stabilization, as well as p21^WAF1^ upregulation compared to RG7388 alone (Figure 4D,E).

There was no obvious correlation between sensitivity to E7107 and SF3B1 status in the cell lines tested, including Nalm-6 isogenic cell lines expressing either endogenous SF3B1^K700E^ or SF3B1^K700K^ (Figure 1A). Western blot was also performed on the isogenic Nalm-6 cells exposed to increasing concentrations of E7107 for 24 h. At doses above IC_50_, the MDM2 protein dramatically disappeared, resulting in dose-dependent p53 protein stabilization. The accumulated p53 protein upregulated its downstream target p21, which at higher doses switched to its intron-retaining isoform (Figure 4F). The effect of RG7388 in combination with E7107 was investigated for Nalm-6 isogenic cell lines using Median-effect analysis. Combined treatment resulted in less synergy or additive effect for Nalm-6 SF3B1^K700E^, depending on drug concentration, compared to the stronger synergistic effect observed for Nalm-6 SF3B1^K700K^ cells (Figure 4G,H).

### 2.5. Combination Treatment with E7107 and RG7388 Increases Cell Cycle Arrest and Apoptosis

Given that the combination of RG7388 with E7107 led to greater inhibition of proliferation compared to either agent alone for TP53^WT^ Nalm-6 and OCI-Ly3 cell lines, FACS analysis after 24 h treatment was carried out to investigate how these effects were reflected in early changes in cell cycle distribution. For OCI-Ly3, treatment with E7107 alone resulted in a significant reduction in S phase (*p* = 0.014) and an increase in G2/M phase (*p* = 0.0045). Treatment with RG7388 alone, however, resulted in a significant reduction in the S phase (*p* = 0.011), with increases in the cell population in G0/G1 phase (*p* = 0.011). E7107 treatment further decreased the proportion of cells in S phase when it was combined with RG7388 (Cells in S phase as % of total events treated with 1xIC_50_ RG7388 alone vs. combination with E7107 *p* = 0.043) (Figure 5A,C).

For Nalm-6 cells, unlike OCI-Ly3, treatment with E7107 alone resulted in a significant increase in G0/G1 phase (*p* = 0.019) and a decrease in the G2/M phase (*p* = 0.014). Treatment with RG7388 alone resulted in a significant increase in G0/G1 phase (*p* < 0.0001). E7107 treatment further decreased the proportion of cells in S phase when it was combined with RG7388 (Cells in S phase as % of total events treated with 1xIC_50_ RG7388 alone vs. combination with E7107 *p* = 0.002) (Figure 5B,D).

The induction of apoptosis was evaluated by caspase 3/7 enzymatic activity, which is a sensitive and specific indicator of apoptosis [22,23]. Wild-type TP53 Nalm-6 and OCI-Ly3 cells were treated for 24 and 48 h with 1 nM E7107 or 0.2 µM RG7388 as single agents and in combination. E7107 treatment alone had no significant effect on caspase 3/7 activity in either cell line, indicating cytotoxicity by a caspase 3/7-independent mechanism (Figure 5E). Combination treatment, however, induced highly significant increases in caspase 3/7 activation at 24 and 48 h time points compared to either DMSO vehicle control treatment or with either agent alone (Figure 5E).

### 2.6. Combined Spliceosome Disruption and MDM2 Inhibition Has a Synergistic Cytotoxic Effect on Primary CLL Cells

The effect of E7107 in combination with RG7388 was investigated for p53 functional primary CLL samples using a dose-response matrix assay. Combination matrix analysis was performed for *n* = 4 CLL patient samples (Appendix A) after incubation with a range of E7107 and/or RG7388 concentrations, focusing on clinically relevant well-tolerated dose ranges. According to recently published early clinical trial data, the plasma concentration peaked at 6000 ng/mL in patients following oral administration, corresponding to approximately 100 µM for idasanutlin [27]. The viability of these CLL samples after 48 h exposure to RG7388 and E7107 was determined as single agents (Figure 6A) and in a matrix combination of doses (Figure 6B). To quantify the degree of synergy, the zero potency interaction (ZIP) reference model [33] was used, visualized as a 3D synergy/antagonism landscape (Figure 6C). Similar to the in vitro human B-cell line results, there was considerable synergy (average ZIP synergy score: 9) across a range of concentrations, even at doses that had very low effects on viability when used alone (Figure 6D). The combined treatment with RG7388 and a low concentration of E7107 (1 nM) resulted in higher p53 stabilisation and apoptosis, as indicated by PARP cleavage (Figure 6E).

## 3. Discussion

Targeted agents have been replacing chemoimmunotherapy in CLL. However, a substantial number of patients still discontinue therapy because of reasons including acquired resistance, the suboptimal durability of response in patients with high-risk disease, and indefinite treatment duration [34]. Therefore, an increasing number of studies are evaluating alternative targeted agents in CLL, including MDM2–p53 binding antagonists. These targeted agents are likely to be most effective in combination with other agents because of the variable response to MDM2 inhibitors and the potential regrowth of any resistant *TP53*^MUT^ subclones. In this study, experiments were conducted to assess the effect of combination treatments with RG7388 and the spliceosome inhibitor E7107 in B cells of known *TP53* status. This was encouraged by the observed disruption of the p53 pathway proteins by E7107 treatment, which, nevertheless, had antiproliferative and cytotoxic effects independent of *TP53* status. Overall, the potential clinical benefit of combining MDM2 inhibitors with splicing modulators was demonstrated in a panel of B-cell lines as well as in CLL primary samples. Here we suggest a novel approach to treat CLL, as well as other haematological malignancies, and have elucidated mechanisms that contribute to the cellular response.

Splicing alterations are a characteristic feature of CLL cells, irrespective of *SF3B1* mutation status, and affect numerous pathways, including, importantly, the p53 pathway [19]. Because the p53 pathway controls a major growth regulatory and death signalling network, altered splicing of p53 pathway genes may have a contributory effect on the cytotoxicity of spliceosome inhibitors. The effect of E7107 was evaluated for key proteins in the p53 pathway. Cell growth was suppressed in a panel of B-cell lines independent of *SF3B1* and *TP53* mutational status, with a production of aberrantly spliced transcripts and protein products of p53 pathway genes. MDM2 protein levels decreased in all cell lines, accompanied by a reciprocal concentration-dependent increase in p53 in *TP53*^WT^ cells. In addition, a lower molecular weight p53 isoform (~45 kDa) was evident in OCI-Ly3 cells treated with E7107, likely due to alternative splicing. Different p53 isoforms were reported in the literature with molecular weights from 26 to 48 kDa, and some of them were shown to exhibit several biological functions, modulating p53 transcriptional activity and tumour-suppressor functions [35,36]. A high molecular weight p21^WAF1^ isoform (~30 kDa) formed due to intron retention was also evident in *TP53*^WT^ cells. Intron retention has been shown as a hallmark of cancer stemness and aggressiveness in multiple studies, and splicing modulation reverses this aggressive behaviour and overcomes treatment resistance in cancers [37,38,39,40]. The detection of these new splice variants in B cells, particularly those of *CDKN1A* and *MDM2*, may provide insights into the studies with splicing inhibitors in terms of drug resistance/synergies and new drug combination strategies. Sequence analysis revealed that despite retaining the N-terminal CDK binding site, the proliferating cell nuclear antigen (PCNA) interaction site and nuclear localization signal domain were partially lost in the intron-retained isoform of *CDKN1A* encoding the p21^L^ protein product. The alteration in the nuclear localization signal of p21^L^ prevents its access to the nucleus, resulting in the loss of both CDK and PCNA inhibitory activity in the cellular context. Ectopic expression of the altered p21 in Nalm-6 cells using lentiviral transduction showed that the p21^L^ protein, with an alternative C-terminus, loses its ability to act as a cyclin-dependent kinase inhibitor due to loss of the ability to localise to the nucleus (Figure 7). The protective effect of normal p21-mediated growth inhibition is lost with the switch to the aberrant p21^L^ isoform that is unable to localize to the nucleus, thus in the context of E7107 treated cells, sensitising them to the concomitant p53-dependent accumulation resulting from the MDM2 downregulation due to exon skipping. Another piece of evidence to support this mechanism is that although isogenic Nalm-6-SF3B1^K700K^ cells exposed to E7107 have the same IC_50_ as SF3B1^K700E^ cells (unpaired *t*-test, *p* = 0.63), they form greater levels of p21 aberrant splicing isoform (p21^L^) at the same concentration of E7107 treatment (Figure 4F). Nalm-6 isogenic cells expressing SF3B1^K700K^, which are likely to be more sensitive to p53-dependent proapoptotic signals due to the aberrant p21^L^ isoform, benefit more from combined treatment with the MDM2–p53 antagonist RG7388, as shown by the increased synergy (Figure 4H).

Growing evidence has shown that defects of RNA processing factors, including spliceosome components, trigger R-loop accumulation and R-loop-associated genomic instability [41,42]. R-loops are an important source of replication stress and genome instability, which are hallmarks of cancer [43]. Although R-loop accumulation alone is not sufficient to promote widespread genomic instability, mis-splicing-associated R-loops potentiate specific oncogenic events [44]. Splicing modulators create a dependence on ATR to suppress DNA damage. ATR inhibitors induced DNA damage and cell death in cells with heterozygous splice gene mutations were enhanced by splicing modulating compounds Pladienolide B and E7107 [45]. These results might be an alternative reason for the synergy between spliceosome and MDM2 inhibitors that we observed in our experiments. In future studies, it would be important to address the contribution of R-loop accumulation to the synergistic effect of these agents.

Combining splicing modulation with RG7388-sensitized primary CLL cells to RG7388 treatment provided a novel strategy to minimize the chance of acquired resistance to MDM2 inhibitors in CLL. The combination treatment and consequent dose reduction might also reduce the reported side effects of E7107 for a limited number of patients during clinical trials [46] by allowing it to be used at lower doses. The combination treatment also provides an advantage in CLL cases where there may be clonal subpopulations of *TP53* mutant CLL cells because the spliceosome inhibitor has a cytotoxic effect independent of *TP53* status. Furthermore, based on the results for E7107, the use of a combined treatment may be an option for other novel small-molecule splicing modulators, such as H3B-8800 [47], which has recently gone into early phase trials (ClinicalTrials.gov Identifier: NCT02841540). Taken together, we identified a novel approach that can be explored further to treat CLL and have uncovered potential mechanisms that contribute to the cellular response. This study provides preclinical data to guide future studies and to improve the outcome for CLL patients and potentially other haematological malignancies.

## 4. Materials and Methods

### 4.1. Cell Lines and Compounds

All of the cell lines (Appendix A) were obtained from authenticated cell line resources and routinely cultured using RPMI-1640 medium (Sigma-Aldrich, St. Louis, MO, USA) with 10% foetal calf serum. E7107 (H3 Biomedicine, Cambridge, MA, USA) and idasanutlin were dissolved in DMSO (Sigma-Aldrich) and used at a final concentration of 0.5% DMSO (*v*/*v*). Idasanutlin was custom synthesised with >99% purity as part of the Newcastle University/Astex Pharmaceuticals Alliance and CRUK Drug Discovery Programme at the Newcastle University Cancer Centre.

### 4.2. Patient Samples

Peripheral blood samples were obtained from CLL patients with informed consent, in accordance with institutional guidelines and the Declaration of Helsinki. CLL patient samples were obtained and stored under the auspices of the Newcastle Biobank (Research Ethics Committee (REC) reference 17/NE/0361). CLL diagnosis was made according to IWCLL-164 NCI 2008 criteria [48]. Appendix A provides the patient criteria included in the study and the details of the *TP53* and *SF3B1* mutations, including coding region position and amino acid changes as well as del17p status.

### 4.3. Cell Viability Assay

#### 4.3.1. Cell Lines

Cells were seeded at 1.6 × 10^5^ cells/mL in 100μL of medium per well of a 96-well plate (Corning) for 24 h before treating with a range of concentrations (from 1 to 10^4^ nM) of E7107 and/or idasanutlin for 72 h. Concentrations are indicated in the figure legends. XTT Assay Kit II (Sigma-Aldrich, Gillingham, UK) was used to measure growth inhibition compared to solvent DMSO control.

#### 4.3.2. PBMCs

5 × 10^6^ cells/mL in 100μL of medium per well of a 96-well were exposed to a range of concentrations (from 1 to 10^4^ nM) of E7107 and/or idasanutlin for 48 h. Concentrations are indicated in the figure legends. Ex vivo cytotoxicity was assessed by XTT Assay Kit II (Sigma-Aldrich, UK). Results were normalised to DMSO controls and expressed as % viability.

#### 4.3.3. Combination Treatment and Median-Effect Analysis

For the combination treatment of idasanutlin and E7107, the wild-type *TP53* cell lines were treated for 72 h with each agent alone and in combination simultaneously at constant 1:1 ratios at 0.25×, 0.5×, 1×, 2×, and 4× of their respective IC_50_ concentrations. Median-effect analysis was used to calculate combination index (CI) values [49] using CalcuSyn software v2 (Biosoft, Cambridge, UK). DMSO solvent concentration used to dissolve drugs was kept constant at 0.5% (*v*/*v*) in final volume for single and combined treatments.

### 4.4. Apoptosis Assay

1.6 × 10^5^ cells/well were seeded in white-walled 96-well plates (Corning, Cat. #3917) and exposed to 1 nM E7107 and/or 0.2 µM idasanutlin for 24 and 48 h. Caspase 3/7 activity was assessed using Caspase-Glo 3/7 Assay (Promega, Southampton, UK). Apoptosis was also determined by examining PARP protein cleavage by Western blot.

### 4.5. Flow Cytometry

0.5 × 10^6^ cells/mL were seeded in 1 mL per well of a 24-well plate (Corning) and exposed to idasanutlin or E7107 alone and in combination at constant 1:1 ratios at 1× of their respective IC_50_ concentrations for 24 h. Appendix A shows the corresponding concentrations of inhibitors used for the combination treatment of the cell lines OCI-Ly3 and Nalm-6. Cells were harvested by washing once with PBS and then fixed with 1 mL 70% ethanol in PBS and stored at 4 °C. Before the FACS analysis, cells were centrifuged at 500 × *g* for 5 min and washed with 1 mL PBS, followed by further centrifugation at 500 × *g* for 5 min, and were then incubated in a 250 µL mixture of DNase-free RNase A (40 µg/mL), propidium iodide (24 µg/mL), and PBS at room temperature for 30 min. Samples were analysed on a FACSCalibur^TM^ flow cytometer using CellQuest Pro software (Becton Dickinson, Oxford, UK). Cyflogic Software Version 1.2.1 (CyFlo Ltd.) was used to manually gate the single cells on the FL2-A vs. FL2-W scatter plot.

### 4.6. Sanger Sequencing

DNA was extracted using a QIAamp DNA Mini Kit (QIAGEN, 51306) according to the manufacturer’s protocol. PCR reactions were run using Platinum^™^ *Taq* Green Hot Start DNA Polymerase (Invitrogen) in a GeneAmp PCR System 9700 thermal cycler (Applied Biosystems, Waltham, MA, USA) using the following primers:*SF3B1 Ex14* F: ACCAACTCATGACTGTCCTTTC, R: ACAACTTACCATGTTCAATG*SF3B1 Ex15-16* F: AACTTAGGTAATGTTGGGGCA, R: TCAACTGACCTGAAATGAAGAGA*SF3B1 Ex18* F: CCTTGGAAAAGCAGTCTAAAAGG, R: GTCAACCTTTTCTAACCACCCA

A PureLink™ PCR Purification Kit (Invitrogen by Thermo Fisher Scientific, Waltham, MA, USA) was used. The purified PCR products were sent to DBS Genomics (Durham University, UK) for Sanger dideoxy sequencing. Sequencing results were analysed by visual inspection of the chromatograms and alignment with a normal reference sequence using Mutation Surveyor DNA Variant Analysis Software (SoftGenetics).

### 4.7. RT-PCR and Sanger Sequencing of CDKN1A and MDM2

Total RNA was extracted using an RNeasy Mini Kit (Qiagen, Hilden, Germany) and cDNA was generated using a High-Capacity cDNA Reverse Transcription Kit (4368814, Thermo Fisher Scientific) as described by the manufacturer. A DNase treatment step with on-column digestion was carried out using an RNase-free DNase kit (Qiagen). PCR reactions were run using Platinum^™^ *Taq* Green Hot Start DNA Polymerase (Invitrogen) in a GeneAmp PCR System 9700 thermal cycler (Applied Biosystems) using the following primers:*CDKN1A* transcript (spanning exons 3 to 4) F: CTGGAGACTCTCAGGGTCGA, R: CTCTTGGAGAAGATCAGCCG*MDM2* transcript (spanning exons 1 to 11) F: CTGGGGAGTCTTGAGGGACC, R: CTTTCACATCTTCTTGGCTGC

The constitutive and altered products, purified from agarose gel, were sequenced by Sanger sequencing.

### 4.8. Immunoblotting

1 × 10^6^ cells/mL (5 × 10^6^ cells/mL for CLL primary cells or healthy PBMCs) were seeded in 2 mL per well of a 6-well plate (Corning) and exposed to a range of concentrations of E7107 and/or idasanutlin. Concentrations are indicated in the figure legends. Protein lysates were harvested using 2% SDS lysis buffer at 24 h, heated at 95 °C for 10 min, and sonicated. Protein concentration was measured using a Pierce™ BCA Protein Assay Kit (Thermo Fisher Scientific, UK). Primary antibodies against p53 (DO-7) (#M7001, Dako), MDM2 (Ab-1) (#OP46, Merck Millipore), p21^WAF1^ (Calbiochem), PARP (Trevigen), SF3B1 (Sigma), Actin (Sigma), Phospho-Rb (Ser780) (Cell Signalling), Phospho-Rb (Ser807/811) (Cell Signalling), Phospho-Rb (T821) (Abcam) and secondary goat anti-mouse/rabbit horseradish peroxidase-conjugated antibodies (Dako) were used. All antibodies were diluted in 5% (*w*/*v*) nonfat milk or BSA in TBS-tween20. Proteins were visualized using enhanced chemiluminescence reagents (GE Healthcare).

### 4.9. Lentiviral Transduction and Gene-Edited Cell Lines

A tetracycline-inducible (Tet-ON) lentiviral expression system, Q3210 plasmid (SnapFast™, Oxford Genetics Limited UK) was used for the stable expression of the gene of interest. The aberrantly spliced transcript form of P21, P21-Long (P21^L^) was inserted into the Q3210 plasmid and then the lentiviral particles were produced by lentiviral packaging in HEK293T host cells. The transduced cell line was induced using 0.5 µg/mL doxycycline (Clontech Labs). Pre-B Nalm-6 isogenic cell lines expressing either endogenous SF3B1^K700K^ or mutant SF3B1^K700E^ were provided by H3 Biomedicine.

### 4.10. Immunofluorescence Microscopy

Suspension cells were fixed on the slides using 100% methanol for immunofluorescence staining with DAPI (VECTASHIELD^®^, Vector Labs) and anti-p21 (Mouse anti-p21^WAF1^ (EA10), OP64, Calbiochem; 1:100 in blocking buffer). Alexa Fluor 488^®^-conjugated anti-mouse IgG (A11001, Thermo Fisher Scientific; 1:500 in blocking buffer) was used as a secondary antibody. A Leica DM6 fully automated widefield fluorescence microscope system with the LAS X software was used to capture images.

### 4.11. Statistical Analysis

Data are presented as mean ± standard error of the mean (SEM) unless otherwise stated. Statistical tests were carried out using GraphPad Prism 6 software and all *p*-values represent paired *t*-tests of at least three independent repeats unless otherwise stated.

### 4.12. SWISS-MODEL

PCNA/p21 structural information for wild-type and altered p21 was identified using SWISS-MODEL workspace, which is an integrated web-based homology-modelling environment (https://swissmodel.expasy.org/, accessed on 1 September 2021). It searches a library of experimental protein structures to determine suitable templates for a given target protein. Based on a sequence alignment between the target protein and the template structure, a 3D model for the target protein is generated [50].

## 5. Conclusions

Overall, our data provide in vitro evidence of the use of RG7388 in combination with E7107, a spliceosome inhibitor that has both p53-independent and p53-dependent effects. The TP53-independent effects of E7107 minimize the risk of selecting *TP53*^MUT^ subclones resistant to MDM2 inhibitors, whereas the p53-dependent effects interact synergistically with MDM2 inhibitors in *TP53*^WT^ cells. This combined treatment strategy provides a novel additional treatment option for patients with p53-functional CLL and potentially other haematological malignancies.

Mechanistically, the notable finding was that splicing modulation with E7107 differentially affected p53 pathway gene splicing, producing exon skipping for MDM2 and consequent activation of p53, while the effect on *CDKN1A* (p21^WAF1^) was intron retention. It was especially noteworthy that the novel p21 isoform lost cell cycle inhibitory activity due to the disruption of motifs for nuclear localisation and CDK binding. The consequent stabilisation of p53 and compromised cyclin-dependent kinase inhibitory activity drives apoptosis and the observed synergy with response to MDM2 inhibitors. A particularly novel aspect of this synergy is the dual inhibition of MDM2; by E7107 at the transcript level and by RG7388 at the protein level, producing greater p53 stabilisation and apoptosis.

## Figures and Tables

**Figure 1 ijms-24-02410-f001:**
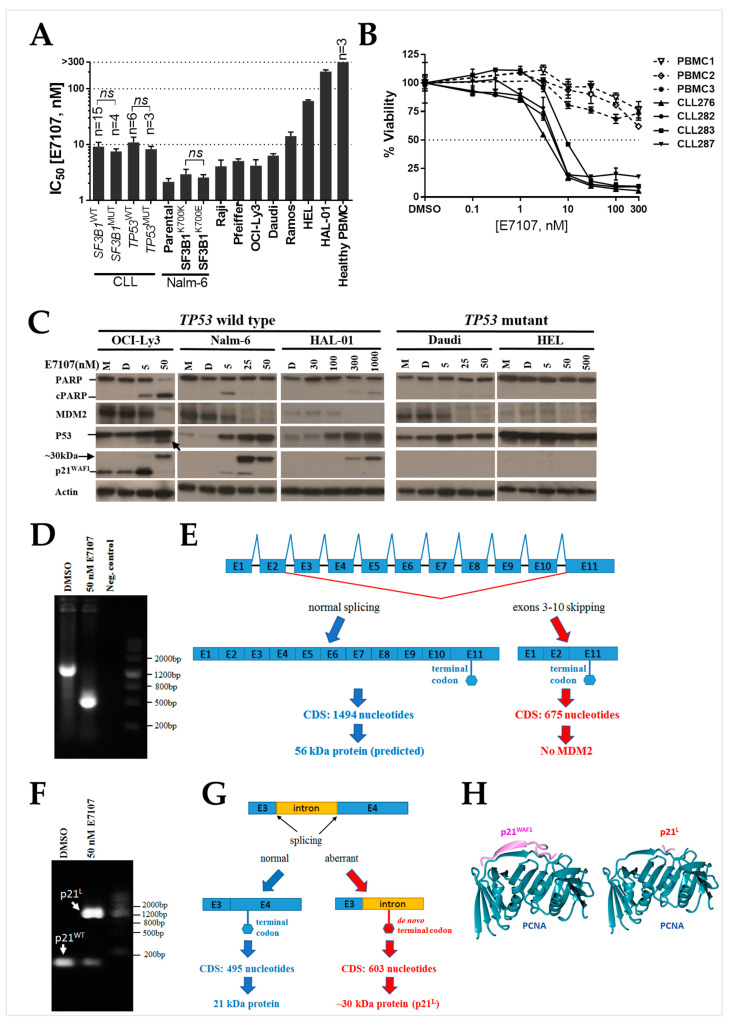
E7107 treatment resulted in aberrantly spliced transcripts and protein products of p53 pathway genes. (**A**) E7107 IC_50_ values for 28 primary CLL patient samples of differing *TP53* and *SF3B1* status and a panel of cell lines, including otherwise isogenic paired *SF3B1* mutant and wild-type Nalm-6 cells. Error bars for the cell lines show the mean ± standard error of the mean (SEM) of at least three independent repeats. (**B**) Dose-dependent inhibition of cell viability using E7107 for four representative CLL primary samples and three PBMC samples from healthy individuals. (**C**) Immunoblot of p53 and its transcriptional targets for *TP53*^WT^ OCI-Ly3, Nalm-6 and HAL-01, and *TP53*^MUT^ Daudi and HEL cells treated with E7107 at concentrations indicated on the figure for 24 h. Actin was used as a loading control. M: No DMSO control cell lysate; D: DMSO solvent control-treated cell lysate. The arrow below the p53 band for OCI-Ly3 points to a lower molecular weight p53 isoform (~45 kDa) likely due to alternative splicing. (**D**) RT-PCR of *MDM2* transcript spanning exons 1 to 11 amplified a shorter product size of approximately 500 bp compared with the expected product size of 1303 bp. OCI-Ly3 cells treated with 50 nM E7107 were compared with untreated DMSO control. Negative control: no template control. (**E**) Schematic diagram of comparison between normal and E7107 altered splicing of *MDM2*. Sanger sequencing confirmed the multiexon skipping of exons 3–10. The skipping of the exons did not change the remaining reading frame and the same terminal codon in exon 11 was maintained. (**F**) RT-PCR of *CDKN1A* transcript using primers spanning exons 3 to 4 revealed the presence of an additional longer transcript product of approximately 1200 bp in OCI-Ly3 cells treated with 50 nM E7107, which was longer than the expected RT-PCR product size of 84 bp. PCR products were analysed on a 2% agarose gel by electrophoresis at 75 V for 45 min. (**G**) Sanger sequencing confirmed retention of the intron between exons 3 and 4, with a de novo termination codon (marked red in the right panel) in the retained intron, which would produce a higher molecular weight protein with an alternative C-terminal amino acid sequence. (**H**) PCNA/p21 structural information. SWISS-MODEL online tool was used. The figure on the left shows the interaction of the part of the p21 protein (139–160) with the PCNA. The figure on the right shows amino acids from 149 to 164 are not present in the altered p21 form with intron retention. Instead, there is a short fragment remaining from the wild-type p21 and a completely new sequence (not shown in the figure).

**Figure 2 ijms-24-02410-f002:**
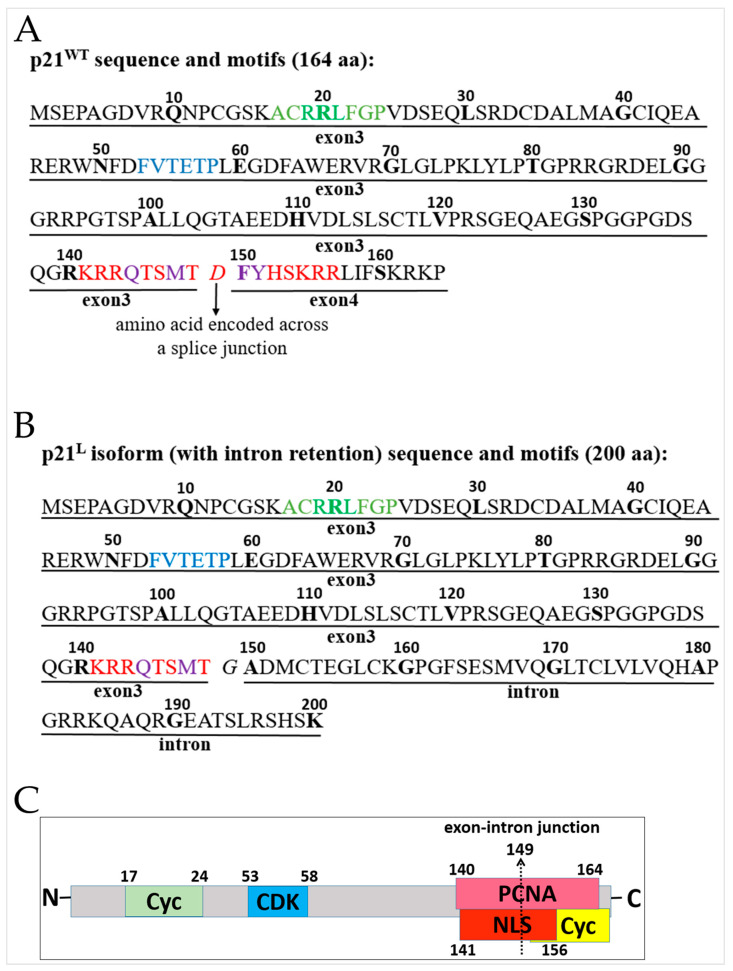
Comparison of the amino acid sequences and colour-highlighted motifs of the wild-type p21 and variant p21^L^ isoforms. (**A**) Wild-type p21; 17–24: RxL motif cyclin binding site; 53–58: CDK binding domain; 140–164: PIP-box K + 4 motif (Qxx[M/L/I]xx[F/Y][F/Y]) proliferating cell nuclear antigen (PCNA) interaction site; 141–156: nuclear localization signal (NLS). (**B**) The p21^L^ intron-retaining isoform of p21. The PCNA interaction site and nuclear localization domain are only partially present in the p21^L^ isoform with intron retention between exons 3 and 4. (**C**) Representation of p21^WAF1^ protein with its protein-interacting domains; 149–200: novel C-terminal sequence.

**Figure 3 ijms-24-02410-f003:**
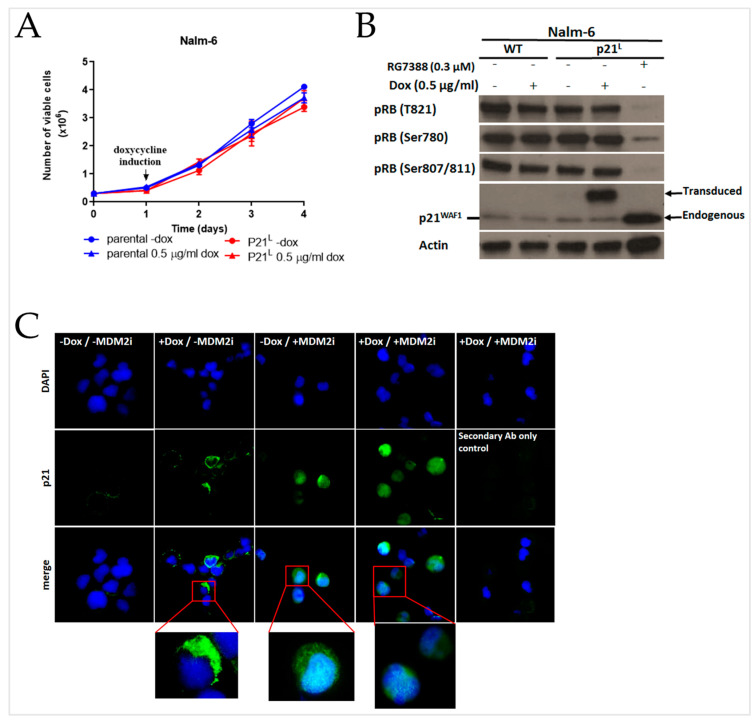
High molecular weight p21^L^ intron-retained isoform with an alternative C-terminus loses its nuclear localisation and consequently its CDK inhibitory activity. (**A**) Growth curves of Nalm-6 parental and Nalm-6–p21^L^ (transduced) cells with 0.5 µg/mL or without doxycycline (-dox) in the medium. The number of viable cells was assessed by trypan blue exclusion. Error bars show the mean ± standard error of the mean (SEM) of three independent counts of viable cells observed under trypan blue exclusion assay. (**B**) Western blots of parental Nalm-6 and Nalm-6–p21^L^ (transduced) either noninduced or induced with 0.5 µg/mL doxycycline for 24 h, showing the expression of the intron-retained variant of p21 in the Nalm-6–p21^L^ transduced cells. MDM2 inhibitor RG7388 (0.3 µM) was used to induce endogenous wild-type p21^WAF1^. RB phosphorylation was suppressed by the induced endogenous p21^WAF1^, but not by the transduced p21^L^ isoform. Actin was used as a loading control. (**C**) The aberrant p21^L^ isoform was unable to localize to the nucleus. RG7388 was used to induce endogenous p21, which showed nuclear localisation. Twenty-four hours after doxycycline and RG7388 administration, cells were fixed on slides for immunofluorescence staining with DAPI (blue fluorescence) and anti-p21. Alexa Fluor 488^®^-conjugated anti-mouse secondary antibody (green fluorescence) was used as a secondary antibody. Fluorescent pictures of each signal, together with the merged images, are shown.

**Figure 4 ijms-24-02410-f004:**
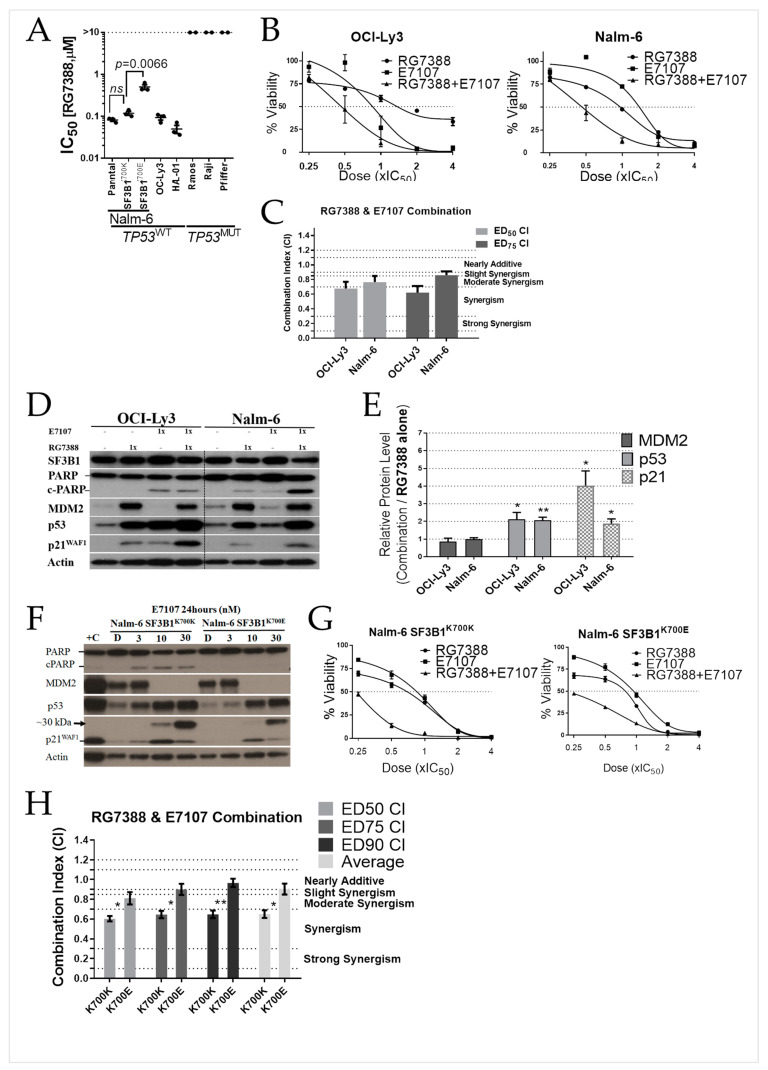
RG7388 synergizes with E7107 for the reduction in TP53^WT^ B-cell line viability. (**A**) IC_50_ values of RG7388 for a panel of TP53 wild-type or mutant B-cell lines. (**B**) Growth inhibition curves for TP53^WT^ B cells treated with RG7388 or E7107, and in combination at constant 1:1 ratios at 0.25×, 0.5×, 1×, 2×, and 4× of their respective IC_50_ concentrations for 72 h. (**C**) CI (combination index) values at ED50 and ED75 percentage dose-effect levels for two cell lines. Data are shown as the mean ± standard error of the mean (SEM) of at least three independent repeats. (**D**) Representative Western blot analysis of SF3B1, p53, p21^WAF1^, MDM2, and PARP/cPARP 24 h after the commencement of treatment with RG7388 or E7107, and in combination at constant 1:1 ratios at 1× of their respective IC_50_ concentration for two TP53^WT^ B cells. Actin is used as a loading control. (**E**) Densitometry analysis of MDM2, p53, and p21^WAF1^ (combination/RG7388 alone) normalized to the loading control, actin. (**F**) Western blot was performed on the cells exposed to increasing concentrations (3, 10, and 30 nM) of E7107 for 24 h. Actin was used as a loading control. D: DMSO treated control cell lysate. +C: Control lysate from OCI-Ly3 cells treated with 0.3 μM RG7388 for 24 h. (**G**) Growth inhibition curves for isogenic SF3B1 mutant and wild-type Nalm-6 cells (SF3B1^K700K^ and SF3B1^K700E^) treated with RG7388 or E7107, and in combination at constant 1:1 ratios at 0.25×, 0.5×, 1×, 2×, and 4× of their respective IC_50_ concentrations for 72 h. (**H**) CI values for RG7388 in combination with E7107 at ED50, ED75 and ED90 for isogenic SF3B1 Nalm-6 cells (wild-type SF3B1^K700K^ and mutant SF3B1^K700E^). Data are shown as the mean ± SEM of at least three independent repeats. Statistically significant differences between the cell lines (* *p* < 0.05; ** *p* < 0.01) are indicated.

**Figure 5 ijms-24-02410-f005:**
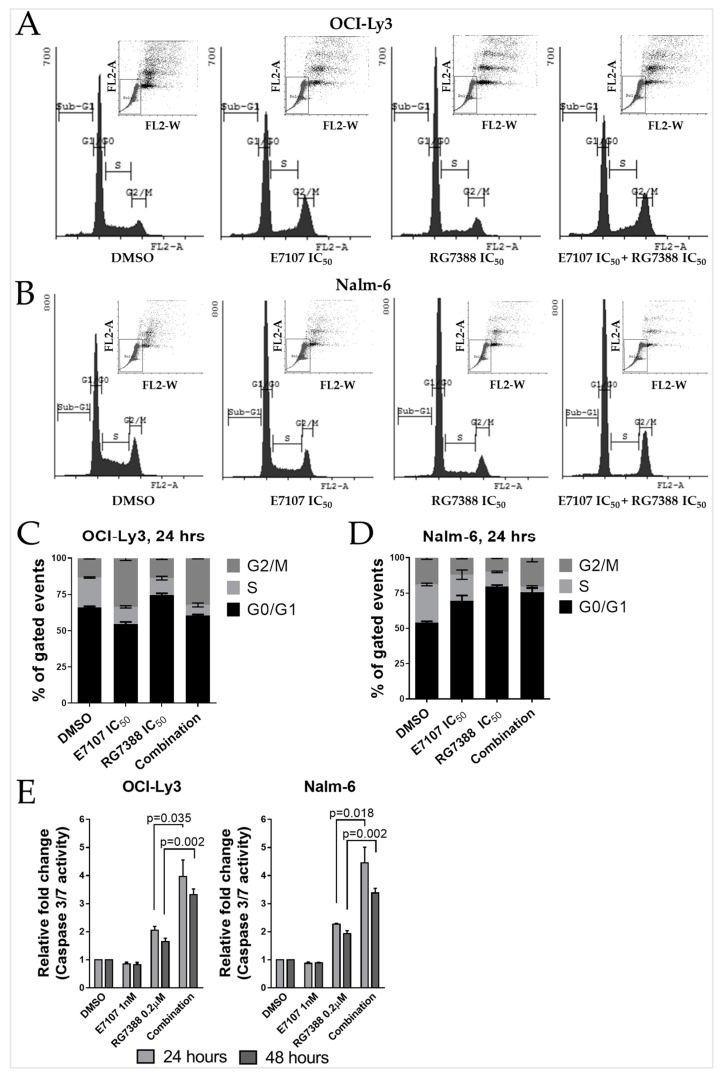
The effect of E7107 on the cell cycle distribution and apoptosis was indicated by caspase 3/7 activity when combined with RG7388. FACS DNA content histograms of TP53^WT^ B-cell lymphoma cell line OCI-Ly3 (**A**) and B-cell precursor leukaemia cell line Nalm-6 (**B**) treated with RG7388 or E7107, and in combination at 1x of their respective IC_50_ concentrations for 24 h. A minimum of 50,000 total cells were analysed per sample. FL2-A vs. FL2-W scatter plots were used to manually gate the single cells as shown in the upper right corner of each histogram. Cell cycle distribution changes by FACS analysis for OCI-Ly3 (**C**) and Nalm-6 (**D**). (**E**) Caspase 3/7 activity as a mechanistic indicator of apoptosis. Caspase 3/7 activity is represented as fold change relative to DMSO solvent control. Data are shown as the mean ± standard error of the mean (SEM) of at least four independent repeats.

**Figure 6 ijms-24-02410-f006:**
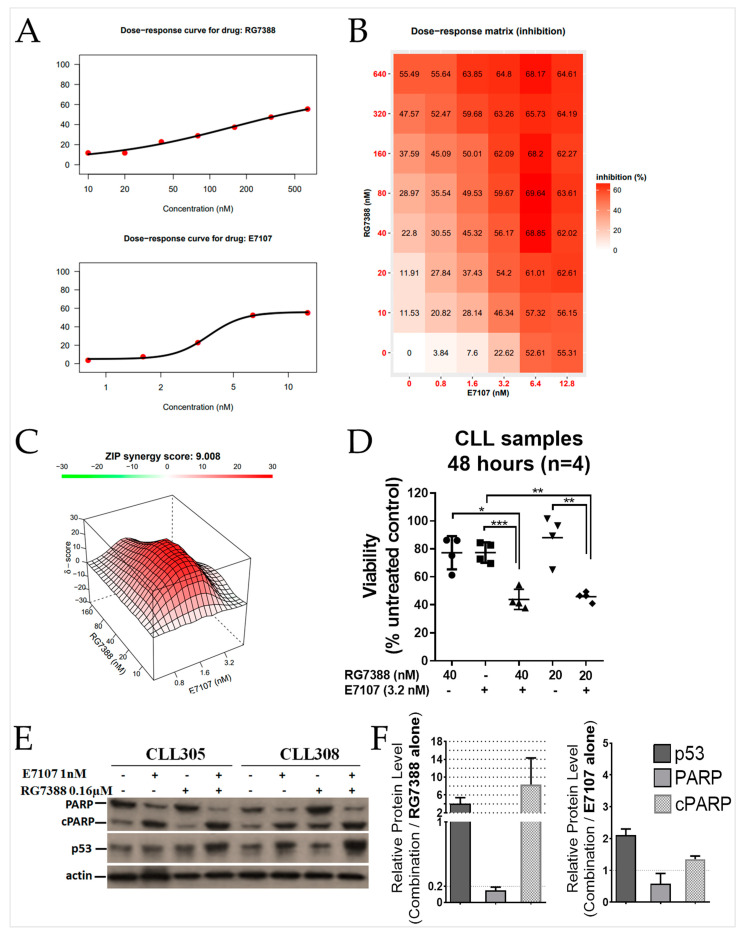
Synergistic effect of RG7388 and E7107 on primary CLL samples. (**A**) Dose-dependent inhibition of cell viability determined by XTT assay averaged for *n* = 4 patient CLL samples after 48 h ex vivo treatment. Curve fitting of the single drug dose-response data was performed using a four-parameter log-logistic function. (**B**) Dose-response matrix heat map showing % inhibition for a range of E7107 and RG7388 dose combinations. (**C**) The synergy heat map for drug combinations using the zero interaction potency (ZIP) model. (**D**) Viability as a percentage of the untreated control for *n* = 4 individual patient CLL samples after 48 h treatment with 3.2 nM E7107 and/or the indicated concentrations of RG7388. Significant differences were calculated by paired *t*-test (* *p* < 0.05; ** *p* < 0.01; *** *p* < 0.001). 20 nM RG7388 alone vs. combination with 3.2 nM E7107 *p* = 0.0091; 3.2 nM E7107 alone vs. combination with 20 nM RG7388 *p* = 0.0051; 40 nM RG7388 alone vs. combination with 3.2 nM E7107 *p* = 0.010; 3.2 nM E7107 alone vs. combination with 40 nM RG7388 *p* = 0.0008. (**E**) Western blot analysis of full-length/cleaved-PARP and p53 proteins 24 h after the commencement of treatment with 0.16 µM RG7388 or 1 nM E7107, or the combination of both for two patient CLL samples (CLL305 and CLL308). (**F**) Densitometry analysis of p53, PARP, and cPARP shows the normalized changes to the loading control, actin. Error bars show the mean with range (*n* = 2).

**Figure 7 ijms-24-02410-f007:**
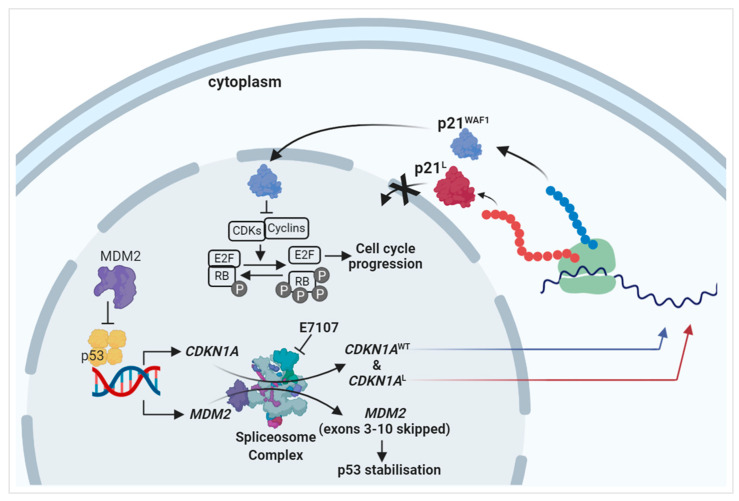
Proposed mechanism for the effect of spliceosome inhibition by E7107 on the p53 pathway and synergy with MDM2 inhibitors. E7107 forces the spliceosome complex to malfunction and produce aberrant products. Reduced full-length *MDM2* production due to exon skipping results in enhanced stabilization and the activation of p53. Nuclear p53 induces more *CDKN1A* transcription, but some of these transcripts are subject to aberrant splicing, in this case in the form of intron retention generating a long form of p21 (p21^L^). Wild-type p21^WAF1^ localises to the nucleus to function as a cyclin-dependent kinase inhibitor to arrest the cell cycle. Altered p21 (p21^L^) protein with an alternative C-terminus, however, loses its ability to act as a cyclin-dependent kinase inhibitor due to the loss of the ability to localise to the nucleus. This loss of the protective effect of normal p21 against apoptosis contributes to the sensitization of the cells to the enhanced p53 stabilisation and p53-dependent proapoptotic signals resulting from the combined effect of MDM2–p53 binding antagonists and *MDM2* exon skipping due to E7107 treatment.

## Data Availability

Not applicable.

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
