# Peer review of "Splicing Modulation Results in Aberrant Isoforms and Protein Products of p53 Pathway Genes and the Sensitization of B Cells to Non-Genotoxic MDM2 Inhibition"

_ijms, 2023, doi:10.3390/ijms24032410_

Round 1
Reviewer 1 Report
In this study the authors have assessed B cell lymphoma anti-tumour activity by the spliceosome inhibitor E7107 in combination with the MDM2 inhibitor RG7388. Combinatorial effects were seen that might be partly related to E7107s ability to disrupt splicing of CDKN1A. That E7107 have previously been withdrawn due to side effects during clinical trial is mentioned by the authors (line 387). Language and statistical analysis are ok.
Find below points to consider:
Line 21-27: Rephrase sentence: “We here…” (does not make sense) and “Splicing modulation…” (too long)
Line 41-43. Rephrase sentence “Multiple…”
Line 80. Advisable to use unpaired ttest when comparing E7107 toxicity to NALM-6 with different SF3B1 mutation status.
Supplemental table 1. Add E7107IC50 values for Nalm-6 SF3B1K700E and K700K.
Line 91 Add ‘mutation’ TP53 ‘mutation’ status.
Figure 1A and B, lines 81, 84. Primary human B cells treated with E7107 should be included together with the PMBCs.
Line 136-140. The method used to determine structural information should be mentioned in the methods section.
Figure 3A. State how many times growth experiments were repeated and add error bars.
Figure 3B. State in the figure whether RG7388 was added in the experiment. Additional data to the figure would otherwise be required; addition of RG73388 to wt and p21L, with and without doxycycline.
Line: 277. State clinically relevant dose range.
Related to figure 7. The authors should comment on the fact that experiments presented in figures 5 and 6 are using concentration of E7017 which did not result in disrupted splicing products (figure 1C). It is therefore unclear whether p21L would contribute to p53 stabilization as proposed by the authors.
Line 404. State purity of RG87388, and how it was dissolved.
Line 412. State whether the study has ethical approval.
Author Response
Thanks for your comprehensive review and provide valuable feedback for the manuscript. We have largely revised this manuscript based on your comments below and hope this manuscript is suitable for publication after revision.
Author's Reply to the Review Report (Reviewer 1)
Line 21-27: Rephrase sentence: “We here…” (does not make sense) and “Splicing modulation…” (too long)
They have been revised and corrected.
Line 41-43. Rephrase sentence “Multiple…”
We have rephrased the sentence.
Line 80. Advisable to use unpaired ttest when comparing E7107 toxicity to NALM-6 with different SF3B1 mutation status.
We have analysed the data again using unpaired ttest (ns, p=0,63).
Supplemental table 1. Add E7107IC50 values for Nalm-6 SF3B1K700E and K700K.
IC50 values were added, and the revised Supplementary Data file was uploaded.
Line 91 Add ‘mutation’ TP53 ‘mutation’ status.
Added.
Figure 1A and B, lines 81, 84. Primary human B cells treated with E7107 should be included together with the PMBCs.
Mononuclear cells were purified by density gradient centrifugation (Lymphoprep, Axis-Shield) following the manufacturer’s protocol. Because the majority of peripheral blood B cells in CLL are malignant CD5+ B cells (Hayes, Busch et al. 2010), PBMCs were used as an ex-vivo model. In Figure-1A and 1B, ‘CLL’ indicates primary human B cells treated with E7107.
Line 136-140. The method used to determine structural information should be mentioned in the methods section.
We have added a new section for this. Please see the new section named ‘4.12. SWISS-MODEL’
Figure 3A. State how many times growth experiments were repeated and add error bars.
We have added the statement below, and error bars as well.
‘Error bars show the mean ± standard error of mean (SEM) of three independent counts of viable cells observed under trypan blue exclusion assay.’
Figure 3B. State in the figure whether RG7388 was added in the experiment. Additional data to the figure would otherwise be required; addition of RG73388 to wt and p21L, with and without doxycycline.
The right side of Figure3 is cut off in the first submitted pdf file when converting from Word file to Pdf file. The orientation of the Figure3 has been corrected and I think the problem is solved now.
Line: 277. State clinically relevant dose range.
We have a added a statement and a reference for this.
Related to figure 7. The authors should comment on the fact that experiments presented in figures 5 and 6 are using concentration of E7017 which did not result in disrupted splicing products (figure 1C). It is therefore unclear whether p21L would contribute to p53 stabilization as proposed by the authors.
As discussed in the discussion section paragraph 2, the increase in p53 stabilization is mainly due to a decrease in the full-lenght MDM2 protein (due to exon skipping). Western experiments also indicate that, MDM2 is more vulnerable to spliceosome modulation by low doses of E7107 (Figure 1C).
Line 404. State purity of RG87388, and how it was dissolved.
We have added these details.
Line 412. State whether the study has ethical approval.
We have added Research Ethics Committee (REC) reference in sections 4.2 and ‘Institutional Review Board Statement’.
Reviewer 2 Report
The authors have described a novel approach to treat Chronic lymphocytic leukemia (CLL) by modulation of the splicing process of key proteins in the p53 signaling pathway.
Here are some comments regarding the article, details of certain methods needs to be specified:
1. In the method section, please mention the compound name rather than the code. For example, Line 402, RG7388.
2. Line 405, Please specify the patient's criteria included in the study. Age, gender, etc.
3. Line 416,421,426,469. Please mention the concentrations, and what were they dissolved in?
4. Line 432, what was the concentration of RG7388 and E7107?
5. Line 471, what was the temperature of the heating process, and for how long?
6. Please provide the conclusion of the study.
7. Line 513, Please provide the ethical clearance certificate number for this study.
8. Figure 4, 5, 6, how was the combination made? Please specify.
9. Please specify the positive controls used in the experiments.
Author Response
Thanks for your comprehensive review and provide valuable feedback for the manuscript. We have largely revised this manuscript based on your comments below and hope this manuscript is suitable for publication after revision.
- In the method section, please mention the compound name rather than the code. For example, Line 402, RG7388.
We have replaced all RG7388's with ‘idesanutlin’ in the method section.
- Line 405, Please specify the patient's criteria included in the study. Age, gender, etc.
Supplementary Table 2 provides the patient's criteria included in the study. A sentence stating this has been added where you specified.
- Line 416,421,426,469. Please mention the concentrations, and what were they dissolved in?
We have added the text below in the ‘4.1 Cell lines and compounds’ section.
‘E7107 (H3 Biomedicine, Cambridge, Massachusetts) and idasanutlin were dissolved in DMSO (Sigma-Aldrich) and used at a final concentration of 0.5% DMSO (v/v).’
Since the concentrations used vary between experiments, the statement 'Concentrations are indicated in the figure legends' was added in text and it was ensured that the concentrations were given clearly in each figure legend.
- Line 432, what was the concentration of RG7388 and E7107?
The statement below was added.
‘Supplementary Table 3 shows the corresponding concentrations of inhibitors used for combination treatment of the cell lines OCI-Ly3 and Nalm-6.’
- Line 471, what was the temperature of the heating process, and for how long?
We have added the temperature and how long it was applied.
- Please provide the conclusion of the study.
We have added a conclusion section.
- Line 513, Please provide the ethical clearance certificate number for this study.
We have added the statement ‘Samples were obtained through the Newcastle Biobank (REC reference 17/NE/0361).’
- Figure 4, 5, 6, how was the combination made? Please specify.
We have added a section explaining combination treatment. Please see the section named ‘4.3.3. Combination Treatment and Median-Effect Analysis’
- Please specify the positive controls used in the experiments.
Positive controls were rechecked.